

# Bridging the gap in medical education: comparing analysis of light microscopy and virtual microscopy in histology

Aysel Başer[1] and Başak Büyük[2]

[1] Department of Medical Education, Faculty of Medicine, Izmir Democracy University, İzmir, Konak, Turkey
[2] Department of Histology and Embryology, Faculty of Medicine, Izmir Democracy University, İzmir, Konak, Turkey

## ABSTRACT

This study aims to investigate the impact of virtual microscopy (VM) and light microscopy (LM) on the satisfaction of second-year medical students and how they affect student performance in different educational settings. The research involved 94 second-year students from Izmir Democracy University's School of Medicine, with criteria requiring enrollment in the 2021–2022 academic year and attendance of at least 80% in histology practical course. A paired two-tailed t-test was used for comparison, with a researcher-designed questionnaire for data collection. Cronbach's alpha was 0.894 for the LM questionnaire, and 0.918 for the VM questionnaire, indicating high level of reliability. LM scored higher in the questionnaire ($p = 0.010$), but VM showed higher exam averages ($p = 0.013$). The study found VM more effective in exams, with students showing high satisfaction with LM. VM's accessibility to histological preparations and its impact on learning levels and board exam success rates were noted. The study concludes that while VM is becoming essential in histology education due to its positive impact on exam performance and accessibility, LM remains highly valued by students for its hands-on experience and satisfaction levels.

## INTRODUCTION

In the evolving landscape of medical education, the integration of digital tools has become increasingly central, particularly in disciplines where visual analysis plays a critical role. Traditional light microscopy (LM), once the cornerstone of histological and pathological studies, is being progressively supplemented or even replaced by virtual microscopy (VM). This transition is driven not only by the need to enhance educational accessibility and efficiency but also by the substantial advancements in digital imaging technologies. Virtual microscopy offers a dynamic platform for examining tissue sections, allowing for detailed digital reproductions that students can access and interact with across various devices. Recent studies, including a comparison of VM and LM in teaching medical histology, highlight VM's potential to significantly improve both student learning experiences and

Corresponding author
Aysel Başer, aysel.baser@idu.edu.tr

academic performance. This shift reflects a broader trend towards integrating more flexible, scalable, and technologically advanced methods in medical training, aiming to better prepare students for a digital-centric healthcare environment (*Krippendorf & Lough, 2005*; *Hattwig et al., 2013*; *Alotaibi & ALQahtani, 2016*).

In studies on the use of visual materials in the education process, it has been revealed that 83% of what is learned is acquired through vision, 11% through hearing, and the rest through other senses. People remember 10% of what they read, 20% of what they hear, 30% of what they see, 50% of what they see and hear, 70% of what they say, and 90% of what they do and say with keeping time constant (*Heinich et al., 1999*; *Seferoğlu, 2009*).

The current rates reveal the importance of visual materials in education. These visual materials used during the training give the student a chance to concretize the information given in the theoretical lessons, simplify the concepts that are difficult to understand, organize the information learned in the theoretical lessons and to be organized by the student, to understand the points that are possible to be missed in the theoretical lessons (*Felten, 2008*; *Hattwig et al., 2013*; *Öz et al., 2021*).

The recent trend in histology education has been incorporating computers into teaching and independent student work (*Harris et al., 2001*). Digital slide imaging technology has provided a unique way to view conventional glass slides electronically in histology laboratories. Using digitalized imaging methods, the images used in the histology practical course are made easily accessible over the internet using a tablet or computer (*Kuo & Leo, 2019*). Due to advances in computing technology, the adoption of virtual microscopy (VM) has transformed traditional practices of laboratory-based light microscopy (LM) use into web-based learning in line with the modern medical curriculum (*Kuo & Leo, 2019*). VM revolutionizes the teaching of histology. Many medical schools have started using this new educational technology, and many are planning to use it (*Dee, 2009*). Recently, the Royal Canadian Medical School of Physicians and Surgeons for anatomical pathologists preferred VM for national licensing certification exams (*Kuo & Leo, 2019*). While VM can provide the image quality provided by traditional light microscopes, it also offers many advantages of technological infrastructures, such as efficiency and accessibility (*Dee, 2009*).

With the integration of VM into the education program, the performance of medical school students has increased. Students and educators have adopted VM as a preferred learning method over traditional LM. This transformation has significantly enhanced learning flexibility by alleviating several constraints associated with LM. Specifically, it has overcome physical limitations of the traditional lab environment, allowed for remote access to slide images, facilitated simultaneous viewing by multiple users, and supported a shift from static to dynamic learning through features like pan-and-zoom control and collaborative annotations. These advancements have extended learning beyond the confines of conventional laboratory settings, embracing a more inclusive and accessible educational framework (*Hattwig et al., 2013*; *Kuo & Leo, 2019*; *Öz et al., 2021*).

Previous studies showed that the performance of learners who use (*Felten, 2008*; *Hattwig et al., 2013*; *Öz et al., 2021*) VM increases significantly, which means that VM can be used as a learning method (*Kuo & Leo, 2019*). This study is strategically planned to understand whether virtual microscopy (VM) genuinely enhances student performance

across different learning styles in medical schools, varying by country, region, and institution. It makes a unique contribution to the field by quantitatively comparing the effects of light microscopy (LM) and virtual microscopy on the exam performance and satisfaction levels of second-year medical students at Izmir Democracy University.

## METHODS

The research was designed in a cross-sectional-descriptive type. It was held at the School of Medicine of Izmir Democracy University (IDU) between September 2020 and December 2021.

### Participants and consent

The students participated in this study voluntarily, and the questionnaires were applied through Google Forms without obtaining their identity information. The survey was administered digitally to the students. Before participating in the survey, an introductory text was displayed to the students, providing them with information about the study. Students who consented to participate in the study by checking the approval box continued with the survey.

Subjects (n:94) were recruited from the second year IDU Medical School students (n:103) at the beginning of their first semester of medical school.

The criteria for inclusion in the research are that students should study at IDU School of Medicine in the 2021–2022 academic year, attend 80% of the histology practical course, and voluntarily participate in the research.

### Educational program

IDU School of Medicine curriculum is planned as vertically and horizontally integrated, organ system-based committees. Every academic year in this medical school is called a term (T), and from the first term, planning has been made by integrating basic cell science into normal human structure and pathology. Each term, five or six committees (C) deal with a particular issue or system. Specifically, Term II comprises six committees, and this study included two of those committees in its scope. The histology laboratory course varies based on the committee, according to the medical school Term II curriculum.

Term 2 Committee 1 (T2C1) at IDU School of medicine is about the Hematopoietic and Immune System, and Term 2 Committee 2 (T2C2) is Circulatory-Respiratory System. The first committee (T2C1) required the identification of cellular organelles from eustatic computer images in histology laboratory studies, and the second committee (T2C2) used light microscopy and slides and required students to examine typical cell structure at the light microscopic level. Slide files were prepared for T2C1 VM images. VM slides of national and international universities and VM images prepared by the lecturer were used to prepare these files. The researchers uploaded VM images for the students to the Microsoft Teams program within the scope of the committee-specific course. Lecture notes containing virtual slides they can view using the computer laboratory within IDU were also shared with the students. In T2C2, LM-based traditional practice lessons were conducted by examining ready-made preparations.
## Measurement tools

Literature review was conducted to develop a form. The measurement tools used in the literature and different previous studies were examined. Using the literature, the researchers have identified titles related to microscope usage experience (*Krippendorf & Lough, 2005*; *Kumar et al., 2004*; *Marchevsky, Relan & Baillie, 2003*; *Munro, 2005*; *Nunnally, 1978*; *Scoville & Buskirk, 2007*; *Wells & Wollack, 2003*). Eighteen items were identified as the observation title. This questionnaire was developed to evaluate the different aspects of the use of virtual microscopy compared to light microscopy from the student's point of view. After the questionnaire was developed by the researchers, feedback was solicited from three additional experts, ensuring representation from six different disciplines in conjunction with the researchers. The experts consulted include an Associate Professor of Histology and Embryology (researcher), an Assistant Professor of Family Medicine and Medical Education (researcher), a Professor of Public Health, an Associate Professor of Dentistry, and an Assistant Professor of Nursing. The questionnaire was reorganized after expert opinions. As a result of the literature review and experiences, a questionnaire consisting of four parts and 40 items (Part 1:2 question+Part 2:18 item LM +Part 3:18 item VM+Part 4:2 open ended question) were prepared by the researchers. Part one; Sociodemographic data (gender and age), part two; (light microscope in the histology laboratory (L1–L18:18 items)) part three (perceptions of different aspects of virtual microscope use performance (V1–V18:18 items)) and part four. Part four of the questionnaire features two open-ended questions. These are designed to gather detailed qualitative feedback from the respondents about the contributions of both Virtual Microscopy (VM) and Light Microscopy (LM) to histology education. Additionally, this section seeks insights into potential areas for improvement within the use of these microscopic techniques in the educational setting. This part aims to capture the direct experiences and suggestions of students, providing a deeper understanding of the practical impacts and possible enhancements needed for both VM and LM in the curriculum. The questionnaire includes statements evaluating students' perceptions of both VM and LM, such as effectiveness, image quality, ease of use, cooperative use, and support.

The second and third sections evaluated 36 statements about using LM and VM using a five-point Likert scale. At the end of the 2021–2022 T2C1 and T2C2 summative evaluation process, the questionnaire developed by the researchers was applied to the students. The students participated in this study voluntarily, and the questionnaires were applied through Google Forms without obtaining their identity information. The survey was administered digitally to the students. Before participating in the survey, an introductory text was displayed to the students, providing them with information about the study. Students who consented to participate in the study by checking the approval box continued with the survey.

## Statistical analysis

In this study, IBM SPSS Statistics for Windows (Version 25.0) was employed to perform the statistical analyses. To assess the central distribution of the data, means and standard deviations were computed. The statistical robustness of the questionnaire's

constructs was tested using the Cronbach's alpha coefficient to determine internal consistency.

To further validate the questionnaire, Kaiser-Meyer-Olkin (KMO) and Bartlett's tests were conducted to ensure the suitability of the data for factor analysis.

For comparative analysis between the LM and VM groups, paired two-tailed t-tests were utilized to identify any significant differences in the mean scores.

### Ethical approval

Ethical approval for the study was granted by the Izmir Democracy University Ethics Committee on 22/07/2020 with the number 2020/17.

## RESULTS

The study involved second-year students from IDU School of Medicine during the 2020–2021 academic year, with a total enrollment of 103. Of these, 91.26% (n:94) of the second-year students (n:103) who took the histology practice course agreed to participate in the study. A total of 54.63% of the students participating in the research were females (n:44).

The analyses of the validity and reliability of the statements in the second and third sections of the questionnaire were made. Cronbach's alpha coefficient was calculated to determine the internal consistency of the questionnaire, this study, in the evaluation of psychometric properties of microscopy items, both LM and VM items were scrutinized. The analysis comprised 18 items for each category, revealing that LM items exhibited a mean score of 72.53 with a variance of 110.25 and a standard deviation of 10.50. When the mean values of the two questionnaires were compared, it was understood that light microscopy had a significantly high score ($p = 0.010$).

Cronbach's Alpha for the LM items was 0.894, and when standardized, it slightly increased to 0.904, demonstrating high internal consistency and reliability of these items in the scale. Cronbach's Alpha for VM items was 0.918, which increased to 0.924 upon standardization, indicating even higher reliability than the LM items, despite the greater spread of scores. These findings suggest that both LM and VM items possess good reliability. These findings underscore the substantial reliability and distinct variability in the assessment of light and virtual microscopy competencies, highlighting the importance of both methodologies in educational contexts (Table 1).

For the light microscopy items, arithmetic mean scores span from a low of 2.88 to a high of 4.47, with an overall average mean score of 4.02. This suggests a generally favorable evaluation among respondents towards light microscopy items. The variability in these scores was highlighted by standard deviations ranging from 0.59 to 1.32 and variances from 0.46 to 1.74, indicating a broad spectrum of responses, with item L14 displaying the lowest mean score and highest variability. Conversely, the virtual microscopy items exhibited a somewhat narrower range of mean scores from 2.53 to 4.26, with an overall average mean score of 3.82. This lower average mean score compared to the light microscopy items suggests a slightly less favorable evaluation. Nonetheless, virtual microscopy items demonstrated greater consistency in responses, as evidenced by

**Table 1 Psychometric properties of light microscopy (LM) and virtual microscopy (VM) items.**

|  | Mean score | Variance | Standard deviation | Cronbach's alpha | Cronbach's alpha based on standardized expressions |
|---|---|---|---|---|---|
| LM item (n: 18) | 72.53 | 110.25 | 10.50 | 0.89 | 0.90 |
| VM item (n: 18) | 68.77 | 163.26 | 12.78 | 0.92 | 0.92 |

**Table 2 Comparative analysis of arithmetic means, standard deviations and variances for light microscopy (L) and virtual microscopy (V) items.**

|  | L1 | L2 | L3 | L4 | L5 | L6 | L7 | L8 | L9 | L10 | L11 | L12 | L13 | L14 | L15 | L16 | L17 | L18 | L_mean |
|---|---|---|---|---|---|---|---|---|---|---|---|---|---|---|---|---|---|---|---|
| Arithmetic mean | 4.04 | 4.41 | 4.07 | 4.11 | 3.96 | 4.19 | 4.07 | 4.18 | 3.90 | 3.96 | 4.30 | 3.39 | 4.01 | 2.88 | 3.80 | 4.22 | 4.32 | 4.47 | 4.02 |
| Standard deviation | 0.88 | 0.68 | 0.92 | 0.85 | 1.23 | 0.91 | 1.02 | 0.93 | 1.04 | 1.05 | 0.84 | 1.31 | 1.00 | 1.32 | 0.99 | 0.81 | 0.85 | 0.71 | 0.59 |
| Variance | 0.77 | 0.46 | 0.84 | 0.72 | 1.53 | 0.82 | 1.04 | 0.86 | 1.08 | 1.09 | 0.71 | 1.70 | 1.00 | 1.74 | 0.98 | 0.65 | 0.71 | 0.51 |  |
| Arithmetic mean | V1 | V2 | V3 | V4 | V5 | V6 | V7 | V8 | V9 | V10 | V11 | V12 | V13 | V14 | V15 | V16 | V17 | V18 | V_mean |
| Standard deviation | 4.01 | 3.99 | 3.97 | 4.02 | 4.22 | 3.56 | 4.04 | 4.18 | 4.15 | 4.17 | 3.39 | 4.26 | 3.87 | 2.53 | 3.93 | 3.21 | 3.60 | 3.66 | 3.82 |
| Variance | 0.97 | 1.00 | 0.98 | 0.95 | 1.08 | 1.22 | 1.07 | 0.95 | 0.97 | 1.05 | 1.22 | 0.87 | 1.18 | 1.31 | 1.08 | 1.36 | 1.20 | 1.19 | 0.71 |
| Arithmetic mean | 0.94 | 1.00 | 0.96 | 0.90 | 1.16 | 1.50 | 1.14 | 0.90 | 0.95 | 1.11 | 1.49 | 0.75 | 1.38 | 1.71 | 1.17 | 1.85 | 1.45 | 1.41 |  |

**Table 3 Results of KMO and Bartlett tests.**

| Results of KMO and bartlett tests |  | LM | VM |
|---|---|---|---|
| KMO | The measure of sampling adequacy | 0.82 | 0.88 |
| Bartlett test | Chi-square | 925.17 | 1255.49 |
|  | df | 153 | 153 |
|  | Sig. | 0.00 | 0.00 |

standard deviations ranging from 0.71 to 1.36 and variances from 0.71 to 1.31, with item V16 showing the greatest variability. The comparative analysis between light and virtual microscopy items reveals nuanced differences in participant evaluations, with light microscopy items generally receiving higher mean scores but also displaying wider variances in certain instances, suggesting varied perceptions and evaluations between these two microscopy techniques (Table 2).

Kaiser-Meyer Olkin (KMO) and Bartlett tests were conducted to examine the sample suitability of the questionnaire. The KMO and Bartlett test results of the study are given in Table 3. According to the survey results, the KMO coefficient of the LM survey was 0.824; the Bartlett test chi-square value was 925.17 (df = 153) and statistically significant (sig. = 0.000). The KMO coefficient of the VM questionnaire was 0.881; Bartlett test chi-square value was found to be 1,255,491 (df = 153) and statistically significant (sig. = 0.000). According to these results, it is a questionnaire with construct validity, normal distribution, and sufficient sampling to perform factor analysis in both sections

(*Munro, 2005*) These analyzes were not carried out because there was no research scale development study (Table 3).

Since the construct validity of both parts of the questionnaire was not similar due to the five significant factors found in the LM statements and the three significant factors in the VM statements, the dependent t-test was applied separately. They have construct validity within themselves, but exploratory and confirmatory analyzes of the factors found in different samples should be made (Table 4).

Table 4 details the results of a dependent t-test analysis comparing students' perceptions and experiences using light microscopy (L) and virtual microscopy (V) across various educational activities, image quality, ease of use, collaborative use, technical support, preference, and overall performance metrics. Statistical results indicate that LM items achieved a higher mean score of 72.53 compared to VM items at 68.77, with a statistically significant difference ($p = 0.010$), suggesting higher satisfaction with LM. Conversely, VM was associated with higher exam scores, indicating its effectiveness in educational performance ($p = 0.013$).

Key findings from this analysis reveal significant differences in several areas:

Total Point and exam performance: When the mean values of the two questionnaires were compared, it was understood that light microscopy had a significantly high score. The overall score comparison between light and virtual microscopy (MD = 0.20, $p = 0.01$) and the specific exam performance comparison (MD = −7.07, $p = 0.01$) demonstrate significant differences in learning outcomes associated with each microscopy method.

Educational activities: Notable differences were observed in how the L/V microscope aligns with course aims and enhances interest in histology. Specifically, Item 2, addressing the alignment with course aims, showed a significant mean difference (MD = 0.43, $p < 0.01$), and Item 6, related to increasing interest in histology, also displayed a substantial mean difference (MD = 0.63, $p < 0.01$). Indicating a stronger alignment and increased interest with either the light or virtual microscopy.

Collaborative use: A significant difference emerged in collaborative use, with Item 11 highlighting an enhanced ability to collaborate (MD = 0.90, $p < 0.01$), suggesting that one format may facilitate better collaboration than the other.

Preference and entertainment: A strong preference for one type of microscopy over the other as a learning tool was evident, as shown in Item 17 (MD = 0.72, $p < 0.01$) and Item 18 (MD = 0.81, $p < 0.01$), indicating a clear favoritism that could impact learning engagement and enjoyment.

Conversely, certain items showed no significant difference, such as the ease of understanding microscope instructions (Item 8, $p = 1.00$) and some aspects of ease of use and technical support, indicating areas where both microscopy types perform similarly from the students' perspective.

When the averages of the two exams were compared, it was found that the average of the exam made with the virtual microscope was significantly higher ($p = 0.01$).

Significant differences were found in 2, 6, 11, 12, 16, 17, 18 items in the LM and VM questionnaire-dependent t-test. There was a significant difference in expressions except for S12; LM averages were high (Table 4).
**Table 4 Dependent t-test results.**

| | Items | | Mean | Standard deviation | Standard error | t | df | Sig. (two-tailed) |
|---|---|---|---|---|---|---|---|---|
| Activity | Item 1: I was able to manage my time effectively using the L/V microscope. | L1–V1 | 0.03 | 1.26 | 0.13 | 0.25 | 93.00 | 0.81 |
| | Item 2: The L/V microscope effectively aligns with the course's aims and learning objectives. | L2–V2 | 0.43 | 1.13 | 0.12 | 3.65 | 93.00 | 0.00 |
| | Item 3: Using an L/V microscope increased the memorability of the subjects. | L3–V3 | 0.11 | 1.22 | 0.13 | 0.84 | 93.00 | 0.40 |
| | Item 4: The L/V microscope helped me learn the subject. | L4–V4 | 0.09 | 1.02 | 0.11 | 0.81 | 93.00 | 0.42 |
| | Item 5: L/V microscopy outside of class hours facilitated my access to information while preparing for the histology practice exam. | L5–V5 | −0.27 | 1.41 | 0.15 | −1.83 | 93.00 | 0.07 |
| | Item 6: Using L/V microscopes in the lesson increased my interest and motivation in histology subjects. | L6–V6 | 0.63 | 1.30 | 0.13 | 4.67 | 93.00 | 0.00 |
| | Item 7: The L/V microscope facilitated my examination of tissues by magnifying them in detail. | L7–V7 | 0.03 | 1.23 | 0.13 | 0.25 | 93.00 | 0.80 |
| Image quality | Item 8: L/V the instructions for using the microscope were clear and concise. | L8–V8 | 0.00 | 0.98 | 0.10 | 0.00 | 93.00 | 1.00 |
| Ease of use | Item 9: The L/V microscope was easy to use. | L9–V9 | −0.24 | 1.30 | 0.13 | −1.82 | 93.00 | 0.07 |
| | Item 10: I can easily navigate images with an L/V microscope. | L10–V10 | −0.21 | 1.30 | 0.13 | −1.58 | 93.00 | 0.12 |
| Collaborative use | Item 11: The L/V microscopy allowed me to collaborate with other students. | L11–V11 | 0.90 | 1.30 | 0.13 | 6.72 | 93.00 | 0.00 |
| | Item 12: Use the L/V microscope outside the scheduled lesson time and any place I want. | L12–V12 | −0.86 | 1.31 | 0.13 | −6.39 | 93.00 | 0.00 |
| Getting technical support | Item 13: When using the microscope during the lesson. it was easier for the trainer to intervene in the sections. | L13–V13 | 0.14 | 1.39 | 0.14 | 0.97 | 93.00 | 0.34 |
| | Item 14: I had trouble browsing the images with the L/V microscope. | L14–V14 | 0.35 | 2.28 | 0.24 | 1.49 | 93.00 | 0.14 |
| | Item 15: Technical problems encountered while using the L/V microscope can be intervened immediately. | L15–V15 | −0.13 | 1.39 | 0.14 | −0.89 | 93.00 | 0.37 |
| | Item 16: There is a greater need for educational support in using L/V microscopes. | L16–V16 | 1.01 | 1.53 | 0.16 | 6.39 | 93.00 | 0.00 |
| Preference/ entertainment status | Item 17: I prefer the L/V microscope as a learning tool in the histology course. | L17–V17 | 0.72 | 1.58 | 0.16 | 4.45 | 93.00 | 0.00 |
| | Item 18: The L/V microscope is fun to use in histology training. | L18–V18 | 0.81 | 1.37 | 0.14 | 5.72 | 93.00 | 0.00 |
| Total point | | L_Mean-V_Mean | 0.20 | 0.72 | 0.07 | 2.63 | 93.00 | 0.01 |
| Committee 1 and 2 exam | | T2C2_LM–T2C1_VM | −7.07 | 27.12 | 2.80 | −2.53 | 93.00 | 0.01 |

"The LM is effective in line with the aims and learning objectives of the course", "The use of LM increased my interest and motivation in histology subjects", "LM enabled me to collaborate with other students", and "Instructional support is more needed in the use of LM", "LM as a learning tool in histology course" I prefer "and" It is fun to use in "LM histology education" were statistically significantly higher in the LM questionnaire ($p < 0.05$) (Table 4).

The statement "I can use the VM outside the planned lesson hours and wherever I want" was found to be statistically significant in the VM questionnaire ($p < 0.05$) (Table 4).

When students were asked about open-ended aspects of LM and VM that needed improvement.

Student 1: Images in the light microscope are catchier.

Student 2: Virtual microscope is effective for learning, but using a light microscope is more enjoyable.

Student 3: Since it is easier to reach virtual microscopic images, we can use it everywhere.

Student 4: The light microscope will contribute more to my work.

Student 5: It may be better to increase the number of light microscopes.

Student 6: The light microscope is more effective for learning, but the virtual microscope is easier. I still prefer the light microscope.

Student 7: The teacher's management of the virtual microscope made the lesson easier.

Student 8: Adding virtual microscope images during lecture processing can increase memorability.

Student 9: Both are effective, but the LM is more effective.

## DISCUSSION

In this study, we evaluated student satisfaction, its impact on learning, and the usability of LM and VM as educational materials through a cross-sectional survey, gathering firsthand insights into students' perceptions of these educational methods. The robustness of our research instruments is underscored by our adherence to rigorous standards of content and face validity, guided by expert evaluations to ensure that our measurement tools effectively capture the intended features of educational methodologies it measures (*Balcı, 2001*; *Bolarinwa, 2015*; *Taber, 2018*).

Statistical reliability analyses, particularly the use of Cronbach's alpha, further strengthen the credibility of our findings. Cronbach's alpha is extensively utilized in educational research to assess the internal consistency of survey instruments, ensuring that the items consistently reflect the construct being measured perceptions (*Balcı, 2001*; *George & Mallery, 2003*; *Bolarinwa, 2015*; *Taber, 2018*). In this study, Cronbach's alpha values for LM and VM were 0.894 and 0.918, respectively, which not only surpass the acceptable threshold of 0.70 but also approach the upper echelons of reliability, indicating a highly reliable scale (*George & Mallery, 2003*). In this study, Cronbach's alpha coefficient for 18 items of the Light Microscope questionnaire was 0.894, and Cronbach's alpha coefficient for 18 items of the Virtual Microscope questionnaire was 0.918. The internal consistency coefficient required from the scales should be over 0.70 (*George & Mallery, 2003*; *Nunnally,*

*1978*; *Schermelleh-Engel, Moosbrugger & Müller, 2003*; *Taber, 2018*). The value we obtained shows that the questionnaire has a highly reliable structure.

To further substantiate the structural validity of our questionnaire, we employed the Kaiser-Meyer Olkin (KMO) measure and Bartlett's test of sphericity to evaluate the appropriateness of our data for factor analysis. Both the LM and VM sections exhibited KMO values well above the 0.70 benchmark, coupled with statistically significant Bartlett's test results, confirming that the data sets were suitable for detailed factor analysis (*Munro, 2005*). This rigorous approach to verifying the questionnaire's construct validity reassures that the instrument is adequately robust for evaluating the nuanced perceptions of students regarding microscopy techniques in medical education.

Despite the strong psychometric properties of our questionnaire and the depth of insights it provided, it is noteworthy that in the broader academic landscape, the use of validated scales in the assessment of educational tools like LM and VM is not commonplace. Most studies do not perform extensive validity and reliability testing on their survey instruments, which may lead to inconsistencies in the evaluation and comparison of educational outcomes across different settings (*Blake, Lavoie & Millette, 2003*; *Sağol et al., 2015*; *Alotaibi & ALQahtani, 2016*).

The findings from this study not only contribute to our understanding of how students perceive and interact with different microscopy techniques but also highlight the critical need for validated instruments in educational research. By establishing a reliable and valid framework for assessing student satisfaction and learning outcomes, future research can build on this foundation to explore more deeply how these tools impact education in the sciences. This could lead to more personalized and effective educational strategies, leveraging the specific advantages of both LM and VM to enhance learning experiences in medical education.

A study conducted with second-year students of the School of Medicine at UCLA School of Medicine, it was shown that the use of VM in pathology practices contributed to the pathology learning of the students, the students were satisfied with using VM, and their learning levels increased. In this way, the participation rates of medical students in pathology courses increased, and the level of course effectiveness increased (*Marchevsky, Relan & Baillie, 2003*). In this study, when the average of the two exams was compared, it was found that the average with VM was significantly higher. Despite the high level of satisfaction with the use of LM, it is thought that the high success rate in the board exam using VM and the easier access to histological preparations with the use of VM increase the learning levels of the students and affect their learning positively. In comparing the findings from the UCLA School of Medicine study with the results of your current study, several interesting contrasts and similarities emerge regarding the use of VM and LM in medical education: Similarly, our study found that VM facilitated higher exam scores, indicating an effective role in learning histology. It is because images in the virtual microscope are a very productive way for students to learn visual material. As stated by the students, VM is easier to access and can be used anywhere. In our study, despite the advantages of VM, such as accessibility and the ability to interact with the digital slides as with a real microscope like as the other articles (*Krippendorf & Lough, 2005*; *Dee, 2009*)

students showed a preference for LM when it came to aligning with the educational objectives of the course. Despite the advantages related to VM, when the mean values of the two questionnaires were compared in our current study, it was understood that the LM had a significant degree score. Considering the items that increase this rate in favor of LM, the students think that the light microscope is more effective than the VM in line with the aims and learning objectives of the course. The reason may be that the students in VM do not spend their practical course hours effectively because they think they can work outside the classroom. Since the students cannot re-evaluate the preparations they see in the LM applications outside of the course time, they reinforce their learning by asking questions to the lecturer during the lesson and discussing with their peers. This may have caused them to find LM applications more effective. Students suggested LM as a factor that increases cooperation with other students in the 11th statement, and this statement was found to be significantly higher in terms of LM.

In the literature, studies such as those conducted at Dokuz Eylül University and the University of South Carolina have primarily utilized surveys to assess the acceptance and efficacy of VM in medical education. These studies report an overwhelming support for VM, noting its ease of use and accessibility as major advantages that potentially enhance learning outcomes and facilitate educational interaction (*Blake, Lavoie & Millette, 2003*; *Sağol et al., 2015*). However, our findings add to the literature by illustrating a significant preference among students for LM when it comes to subjective satisfaction. This preference might be attributed to the tactile and direct nature of LM, which could engage students more deeply by providing a hands-on experience that VM lacks. The tactile feedback and direct manipulation of physical slides in LM might offer cognitive and educational benefits not fully replicable by VM, suggesting a potential area for further research into how these aspects affect learning and satisfaction.

Furthermore, while previous studies have demonstrated VM's capacity to standardize learning materials and potentially reduce logistical burdens, they have not extensively explored the impact of these changes on students' subjective experiences with the microscopy techniques (*Blake, Lavoie & Millette, 2003*; *Sağol et al., 2015*; *Alotaibi & ALQahtani, 2016*). Our study fills this gap by highlighting that despite VM's practical benefits, a significant number of students still show a strong preference for the traditional LM approach.

Another item in favor of LM in the study is that students see using LM in the course as a factor that increases their curiosity and motivation in histology subjects. This may be because students feel closer to their profession while using LM. As the tools such as computers, tablets, and smartphones used during VM are not specific to medicine, they are tools that everyone can use. However, the use of LM and the ability to evaluate tissue in LM is a medicine-specific skill. Therefore, the use of LM can be a factor that increases students' curiosity and motivation.

According to the survey, it was seen that the students need the trainer's support in using LM significantly more than the VM. This may be because students are familiar with the technological tools they use during VM use. Because most of the students encounter LM for the first time in the School of Medicine. For this reason, they may have difficulties even

in the stages before evaluating the histological preparations. Even the preparation placement, the appropriate lens selection, and the ability to sharpen the image and change the area can be difficult for students. Since they received one-to-one support from the trainer at all these stages, this item may be significantly higher in the LM direction. The students may have preferred LM as the preferred method as a learning tool in Statement 27 due to the one-to-one support they received from the trainers during the use of LM and, therefore, the one-to-one communication they established.

In VM, students use technological devices such as computers, which they are familiar with in their previous education. However, they use LM professionally in the School of Medicine for the first time. This makes the LM an object to be discovered and learned for them. For this reason, in statement 18, "it is fun to use the microscope in histology education." The results of the item may have found a significant height in the LM direction.

VM provides the convenience of accessing histological tissue sections from any computer, tablet, or smartphone (*Dee, 2009*). With this aspect, it is easy for students to study histology preparations at any time and place they want. In this study, the result of the 12th statement, "I can use it with a microscope, outside the planned lesson hours and wherever I want" was found to be significantly higher in favor of VM.

Although some medical schools have abandoned the microscope in favor of computer programs, students are concerned about using this technology alone (*Blake, Lavoie & Millette, 2003*; *Krippendorf & Lough, 2005*). It is seen that this concern is primarily for the future that is because they think that they will contribute more to the students in their professional lives.

Although it is a controversial issue for medical students to learn to use LM, it would be appropriate for the institutions that set the standards of medical education at the national level to decide. In the Pre-Graduate Medical Education National Core Education Program 2020, skills are included in basic medical practices such as performing and evaluating complete urine analysis (including microscopic examination), preparing stool smears and microscopic examination, and using a microscope. These skills include performing and evaluating complete urinalysis (including microscopic examination), preparing stool smears, performing a microscopic examination, and using a microscope. After graduating in Turkey, most physicians, especially those working in primary care, do not use the microscope especially to look at histological preparations. However, many primary care physicians use a microscope to look at urine sediments, stool, and blood smears. Students need to learn to use the microscope in the context of learning to look at these examples. Most computer programs are limited to static images that are not functionally microscope-like, as they do not allow students to explore relationships, identify structures independently, and explore relationships by moving the tissue and changing the magnification. Still, static images in textbooks and on the computer, screen undoubtedly enhance the laboratory experience. However, many primary science educators believe that viewing slides of human tissue under a light microscope adds an entirely different dimension to learning that is not provided by still/static images.

Based on formative evaluation, the VM Lab appears to be a viable addition, though not a replacement for the LM and glass slides. Students scored significantly higher on the accessibility and efficiency of the VM Lab compared to the LM lab. In addition, students rated the VM's image quality and navigation as equal to or better than an LM. Virtual slides are always in focus with the ideal condenser and light setting, thus reducing student time and the frustration of running an LM (*Harris et al., 2001*). Our study clearly stated that using VM was more effective in open-ended questions, although the students gave high scores for LM to the survey questions. This shows that students want to use LM and know its effectiveness but expect it to contribute to the lessons without giving up on VM in terms of supporting education.

In light of these observations, it becomes apparent that educational institutions should consider maintaining a hybrid approach that incorporates both VM and LM. Such an approach would not only leverage the technological benefits of VM but also preserve the engagement and satisfaction associated with LM. This balanced approach could cater to diverse learning preferences and potentially enhance overall educational outcomes in medical histology courses. This discussion underscores the complexity of integrating new technologies into traditional educational settings. It highlights the importance of not only considering technological advancements and their potential to improve educational efficiency but also understanding and addressing student preferences and the nuances of educational engagement. Future research should continue to explore these dynamics, particularly through longitudinal studies that could provide deeper insights into how these preferences impact long-term learning outcomes.

## CONCLUSION

This study uniquely contributes to the field by quantitatively comparing the impact of LM and VM on second-year medical students' exam performance and satisfaction levels at Izmir Democracy University. Although the data collection tool used in our study may require testing the scale in different conditions with larger and diverse sample groups from various healthcare professions, it remains a valid and reliable scale for evaluating the effectiveness and satisfaction levels of LM and VM in medical education institutions. Unlike previous research that broadly asserts VM's superiority, our findings provide a nuanced perspective by demonstrating that while VM leads to higher exam scores, students continue to show a strong preference for LM, suggesting a need for balanced integration of both technologies to cater to diverse learning styles and enhance overall educational efficacy in histology. Designed to evaluate students' perceptions of various aspects of using LM and VM for practical applications in histology laboratory courses in medical education, this research provides insights into the tailored integration of both methods in the curriculum.

## LIMITATIONS

The fact that the research was conducted in only one public university is a limitation in terms of being unable to evaluate the changes that may arise from the difference in institutions and that foundations and other public universities are excluded from the

sample. In addition, the questionnaire was applied online in order not to create a risky environment due to the pandemic. An additional limitation regarding the cost-effectiveness analysis in this study could be the lack of comprehensive cost data comparing the initial setup and ongoing operational expenses of VM and LM systems.

### Funding
The study received no funding.

### Competing Interests
The authors declare that they have no competing interests.

### Author Contributions
- Aysel Başer conceived and designed the experiments, performed the experiments, analyzed the data, prepared figures and/or tables, authored or reviewed drafts of the article, and approved the final draft.
- Başak Büyük conceived and designed the experiments, performed the experiments, authored or reviewed drafts of the article, and approved the final draft.

### Human Ethics
The following information was supplied relating to ethical approvals (*i.e.*, approving body and any reference numbers):

Ethical approval for the study was granted by the Izmir Democracy University Ethics Committee on 22/07/2020 with the number 2020/17.

### Data Availability
The raw data are available in the Supplemental Files.

### Supplemental Information
Supplemental information for this article can be found online at http://dx.doi.org/10.7717/peerj.17695#supplemental-information.

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
