# Peer review of "Bridging the gap in medical education: comparing analysis of light microscopy and virtual microscopy in histology"

_PeerJ, doi:10.7717/peerj.17695_

## Round 0.1 · original submission · Minor Revisions

Dear Authors,

As per the observations made by our reviewers, the manuscript has few points to be addressed. They are mention in the attached annotated documents. Please do the needful and submit your revision asap.

All the best

Reviewer 1 ·

Basic reporting

No comment

Experimental design

no comment

Validity of the findings

no comment

Additional comments

Corrections are marked in the manuscript

Annotated reviews are not available for download in order to protect the identity of reviewers who chose to remain anonymous.

·

Basic reporting

Manuscript is structured well but there are some typological and errors. At some places reframing of sentences is required.
 Coherence is missing at many places in the manuscript.
 Content & figures are relevant & raw data is provided.

Experimental design

Original research,
Needs more clear definitions. All the statistical methods/tests used in study have not been stated under statistical analysis e.g. KMO & Barlett test, chronbach alpha coefficient.

Validity of the findings

Results are tabulated and legends are provided, however author is suggested to provide inference of results after each table as tables are not self-explanatory to the reader.

Additional comments

Purpose of the study is missing in abstract.
 Line no. 32- 36; the stanza starts with ‘the current rates….’ But the references stated are more than 20-year-old (1994) and 30-year-old (1984) which definitely not in line with the statement or the current scenario. Author is advised to cite recent references which are not more than 10-year-old.
 Line 53- 59- has been taken from single author’s work. More references should be added to justify these statements.
 Line 84- the context of this particular line is not clear, what actually author wants to state.
 Line no.102 – the citation is from 1978’s era and is it is not available in the reference list.
 Line 105- 108- the statement made here is not clear about opinion from 3 experts (while stated more than 3 experts) or opinion on 3 different aspects was taken from the stated experts & if so, what are the aspects focused on or kind of opinion was sought.
 Line no. 109 & 110 needs revision as the same information is repeated.
 Line no 298- 360- no references cited or any comparison with previous studies provided with the current results.
 Discussion needs to be REWRITTEN as no proper format of discussion has been followed. It seems more like authors personal viewpoint elaborated without providing sufficient references.
 There are so many references cited in-text but not available in list provided e.g. in line no 31, 36, 102, 103, 248, 259,
 Also, there are so many references provided in the list of references at the end but nowhere cited in the text eg. At line no. 392, 394, 397, 409, 427, 432.

Reviewer 3 ·

Basic reporting

The suggested title is “Bridging the gap in medical education: Comparing light microscopy and virtual microscopy in histology.” A small paragraph on overview of microscopy should be incorporated in the beginning of introduction instead of discussing about the case study directly. The text is written in a good and comprehensive English. Structure conforms to PeerJ standards. Tables are well structured. Elaborate line-110.

Experimental design

Research findings are novel. Material and methods are well explained and informative to replicate. Original primary research within the scope of the journal. Research question well defined, relevant & meaningful. It is stated how the research fills an identified knowledge gap. Rigorous investigation performed to a high technical & ethical standard. However, explain why the number of statements were only 36 and the experimental groups were 94?

Validity of the findings

Novelty of the present work has been assessed. The literature is well referenced and also underlying data have been provided to support the findings. The conclusion part is well explained. Statistical findings are also valid.

Additional comments

Nil

Annotated reviews are not available for download in order to protect the identity of reviewers who chose to remain anonymous.

·

Basic reporting

The manuscript is written nicely and presented well.

Experimental design

The experimental design is good, but there is a need to include more people from various age groups to enhance the evaluation and authenticity of the results.

Validity of the findings

The study's findings are currently limited to the included age group, and additional analysis is warranted. Increasing the size and number of participants, if possible, could provide a more comprehensive understanding of the topic under investigation.

Additional comments

1. A notable concern is that only one specific age group is included. If the number of age groups is expanded, the study's validity and authenticity will be enhanced. Increasing the diversity of age groups can provide a more robust understanding of the topic under investigation.
2. The area of learning and its examination is always difficult to analyze and interpret. In that sense, it's a very good effort to tackle such complexities.

---

## Round 0.2 · Minor Revisions

Dear Authors, all the reviewers have checked the submission and appreciate your sincere efforts to address the previously raised points and to improve the manuscript. However, few minot points are still to be checked and addressed. Therefore, I invite you for minor revision of the manuscript. Please make all the desired corrections/justify your own stands, and submit asap.
All the best.

Reviewer 1 ·

Basic reporting

Already reviewed the article. Few corrections are there which are marked in the manuscript

Experimental design

No comment

Validity of the findings

No comment

Additional comments

Corrections are there which are marked in the manuscript

Annotated reviews are not available for download in order to protect the identity of reviewers who chose to remain anonymous.

·

Basic reporting

Clear and unambiguous, professional English used throughout.

Literature references, sufficient field background/context provided.

Experimental design

Original primary research within Aims and Scope of the journal.
Research question well defined, relevant & meaningful.

Validity of the findings

Meaningful replication encouraged where rationale & benefit to literature is clearly stated.

Reviewer 3 ·

Basic reporting

I would like to thank the authors for addressing my comments and all the necessary changes were incorporated in the manuscript.

Experimental design

All the necessary changes were incorporated in the manuscript.

Validity of the findings

All the necessary changes were incorporated in the manuscript.

Additional comments

The authors have provided a nicely detailed and thorough response to the comments from the previous review and have addressed my major concerns. The authors have sufficiently improved their paper, in reaction to the comments made.

---

## Round 0.3 · accepted · Accept

Dear Authors,

As per the recommendations of our reviewers, the manuscript is accepted for publication in PeerJ. This is an editorial acceptance and few more tasks are to be completed prior to publication.

All the best for your future submissions.

Reviewer 1 ·

Basic reporting

no comment

Experimental design

no comment

Validity of the findings

no comment

Additional comments

Already reviewed this manuscript earlier.
Author has incorporated all the corrections in the manuscript